# RecurScan: Detecting Recurring Vulnerabilities in PHP Web Applications

## ABSTRACT

Detecting recurring vulnerabilities has become a popular means of static vulnerability detection in recent years because they do not require labor-intensive vulnerability modeling. Recently, a body of work, with HiddenCPG as a representative, has redefined the problem of statically identifying recurring vulnerabilities as the subgraph isomorphism problem. More specifically, these approaches represent known vulnerable code as graph-based structures (e.g., PDG or CPG), and then identify subgraphs within target applications that match the vulnerable graphs. However, since these methods are highly sensitive to changes in the code graph, they may miss a significant number of recurring vulnerabilities with slight code differences from known vulnerabilities.

In this paper, we propose a novel approach, namely RECURSCAN, which can accurately detect recurring vulnerabilities with resilience to code differences. To achieve this goal, RECURSCAN works around security patches and symbolic tracking techniques, detecting recurring vulnerabilities by comparing symbolic expressions and selective constraints between the target applications and known vulnerabilities. Benefiting from this design, RECURSCAN can tolerate the code differences arising from complex data or control flows within the applications. We evaluated RECURSCAN on 200 popular PHP web applications using 184 known vulnerability patches. The results demonstrate that RECURSCAN discovered 232 previously unknown vulnerabilities, 89 of which were assigned CVE identifiers, outperforming state-of-the-art approach (i.e., HiddenCPG) by 25.98% in precision and 87.09% in recall.

## 1 INTRODUCTION

Over the past decade, PHP web applications have become an integral part of people's daily lives. According to the statistics [4], over 79.2% of online websites are developed using PHP, including many well-known applications such as Facebook [5] and Spotify [7]. However, when providing various useful services, PHP web applications are also exposed to significant security risks. It is reported that, approximately, every 39 seconds, an attack against web applications occurs [3]. Even worse, it is predicted that by 2025, the damages caused by web attacks will reach a staggering 10.5 trillion USD [2].

To safeguard websites from attacks, static analysis techniques have been commonly used to detect vulnerabilities in web applications [11, 13–15, 20, 27, 29, 39, 42]. In particular, vulnerabilities are reported when untrusted inputs (a.k.a, sources) have undergone sanitization along the paths leading to security-sensitive functions (a.k.a, sinks). However, these works often require an accurate modeling of the faulty sanitization logic for different vulnerability types, which requires significant expert experience and is prone to errors (e.g., incorrect sanitization modeling [25, 43, 44]).

To mitigate these limitations, previous works [19, 21, 24, 26, 28, 44, 45] have introduced an alternative approach. Considering that vulnerable code frequently propagates due to the common programming habits of developers, such as copy-and-paste programming or making similar coding mistakes, this line of work recasts the problem of static vulnerability identification as recurring vulnerability detection. Based on this idea, these approaches automatically extract various vulnerability causes from numerous known vulnerabilities and then discover recurring vulnerabilities by detecting the presence of extracted causes in target applications. In this way, these approaches sidestep the labor-intensive task of manually modeling while achieving good accuracy.

HiddenCPG [44], as a representative work on recurring vulnerability detection for PHP web applications, has successfully identified numerous severe vulnerabilities. HiddenCPG first converts known vulnerabilities into code property graphs ($CPG_{\text{vuln}}$) by static anlaysis [11]. Then it detects recurring vulnerabilities by strictly requiring the CPG of the target program to include an isomorphism subgraph of $CPG_{\text{vuln}}$. Nevertheless, HiddenCPG might miss a significant number of genuine recurring vulnerabilities due to its strict isomorphic subgraph matching approach. The underlying reason is that *the CPG-based matching is highly sensitive to code differences*. Usually, after copying the vulnerable code, the developer also makes minor adjustments to fit the context of the target application. Under these circumstances, despite two pieces of code sharing identical vulnerability causes, the slight code differences between them will change their code graphs at various levels (e.g., the numbers of nodes and edges), leading to failures in isomorphic subgraph matching. Due to this limitation, HiddenCPG only achieved a recall of 53.45% in our ground truth, which consists of 232 real-world vulnerabilities.

In light of this, we aim to build a system to accurately detect recurring vulnerabilities in PHP web applications. The most significant challenge to achieve this goal is *how to tolerate the code differences between known vulnerabilities and similar vulnerable code*. On one hand, the copied code may change the implementations of the vulnerable data flow. Thus, the ideal recurring vulnerability detection should tolerate the different implementations of the same vulnerable data flow. On the other hand, the copied code may introduce new conditional statements, some of which perform security checks while others serve for business logic. Ignoring newly introduced security checks may falsely report protected vulnerable code as vulnerabilities. Therefore, the ideal recurring vulnerability detection should be sensitive to the newly introduced security-related constraints while not being corrupted by irrelevant constraints.

In this paper, we propose RECURSCAN, a novel approach for recurring vulnerability detection. Similar to most existing approaches, RECURSCAN first generates signatures for known vulnerabilities and then performs recurring vulnerability detection. To tackle the above problems, RECURSCAN features two new techniques, i.e., symbolic vulnerable data flow matching and selective safe control flow checking. First, inspired by existing researches [35, 46, 47], RECURSCAN calculates the symbolic expressions of sink parameters via symbolic

tracking to represent the data flow cause of known vulnerabilities. When detecting recurring vulnerabilities, RECURSCAN locates potential vulnerability by matching the symbolic expressions for the sinks of the same vulnerability type in the target applications. With the symbol-based matching, RECURSCAN can match the same vulnerable data flow under different implementations. Second, given that the constraints introduced by security patches are typically designed for vulnerability fixing, RECURSCAN extracts safe constraints by analyzing the patch modifications. When discovering a potential vulnerability in the target applications, RECURSCAN further inspects whether the vulnerable code has been protected by safe constraints to avoid false positives. This selective checking policy allows RECURSCAN to avoid interference from security-independent constraints in the target program. In summary, these two new techniques enable RECURSCAN to tackle the code differences between the vulnerable version and the target version, which improves the effectiveness of recurring vulnerability detection.

We implemented a prototype of RECURSCAN targeting injection-based vulnerabilities in PHP web applications. Before conducting the detection, RECURSCAN first automatically constructed a signature database from 184 known vulnerability patches, which includes 249 vulnerable expressions and 27 safe constraints. We then applied RECURSCAN to 200 popular PHP web applications to evaluate its effectiveness. It turns out that RECURSCAN successfully identified 232 vulnerabilities with only 19 false positives. As of now, we have received 89 CVE identifiers. In addition, we compared RECURSCAN with the state-of-the-art approach, HiddenCPG. The results demonstrate that RECURSCAN outperforms HiddenCPG by 25.98% in precision and 87.09% in recall.

In summary, we make the following contributions in this paper:

- We propose a novel approach that can accurately detect recurring vulnerabilities with resilience to code differences.
- We implemented a prototype of RECURSCAN and evaluated its effectiveness in 200 popular PHP web applications. As a result, we found 232 vulnerabilities with 89 CVE identifiers assigned.
- We compare RECURSCAN with HiddenCPG, and the results demonstrate that RECURSCAN outperforms the state-of-the-art approach by 25.98% in precision and 87.09% in recall.

## 2 OVERVIEW

### 2.1 Problem Statement

Given that vulnerable code often propagates across applications, many works [19, 21, 23, 24, 34, 44, 45] have been devoted to detecting recurring vulnerabilities via static analysis. HiddenCPG [44] is the state-of-the-art approach which aims to detect recurring vulnerabilities in PHP web applications. The core idea of HiddenCPG is to extract vulnerability causes from known vulnerabilities and to detect vulnerability by solving an isomorphism subgraph matching problem. Benefiting from its novel design, HiddenCPG does not require manual modeling of vulnerability causes, which is a great challenge for traditional static analysis. Specifically, for a given vulnerable code snippet, HiddenCPG transforms it into a code property graph ($CPG_{vuln}$). During the vulnerability detection, HiddenCPG strictly requires the code property graph of the target program to include the isomorphic subgraphs of $CPG_{vuln}$ to determine the presence of recurring vulnerabilities in the target program.

**Limitations:** Although HiddenCPG detects many severe vulnerabilities with good precision, we found that it still has great limitations and might miss a large number of recurring vulnerabilities. In particular, HiddenCPG's detection strategy requires that unknown vulnerabilities have CPG subgraphs that strictly match those of known vulnerabilities. Apparently, this detection strategy is very sensitive to code changes that affect the CPG graph. That is, any minor code difference between the known vulnerability and a recurring vulnerability will cause the isomorphic subgraph matching to fail, resulting in HiddenCPG missing the recurring vulnerability. Unfortunately, developers often make some adjustments after copying the vulnerable code, thus rendering HiddenCPG ineffective.

**Examples:** We take two typical scenarios of code adjustments as examples to further specify the limitations of HiddenCPG. Figure 1 (a) depicts a piece of PHP code with an XSS vulnerability. This code permits the untrusted variable $id to be directly embedded into HTML content without any form of sanitization or validation, allowing attackers to inject malicious JavaScript code and exploit the vulnerability. Figure 1 (b) and (c) show the code snippets that share the exactly same vulnerability logic as (a) but with slight code differences in data-flow assignments (line 3 in (b)) or irrelevant control-flow statements (lines 3-5 in (c)). When we input (a) as the known vulnerable code snippet into HiddenCPG, it fails to match the recurring vulnerability in (b) and (c) because HiddenCPG is poorly tolerant of CPG changes. The detailed reasons are as follows:

❶ *Difference introduced by different assignments.* It is usual that the sink parameters with the same vulnerable value ranges may have different assignment statements in various programs, resulting in significant differences in the code property graphs. As illustrated in Figure 1 (a) and (b), the former developers chose to directly concatenate $id with strings and output it into the HTML content, while the latter assigned the entire value to the $content variable before outputting it. However, even these minor code differences can lead to entirely different code property graphs: the simplified code property graph of (b) contains more nodes and edges compared to (a). These differences make HiddenCPG struggle to match isomorphic subgraphs and miss obvious recurring vulnerabilities.

❷ *Difference introduced by irrelevant control-flow statements.* After copying the vulnerable code, developers usually insert some new statements that change the control flow. Even if these statements are unrelated to the vulnerability, they would lead to significant differences in the code property graphs. As shown in Figure 1 (c), the code snippet has exactly the same vulnerable code as (a), without any differences. However, HiddenCPG fails to match the vulnerability due to the code differences that result from irrelevant control-flow statements in lines 3-5. In fact, lines 3-5 implement an if-conditional block, which outputs a specific message when the user inputs the value of $id as "0". These lines of code should not be considered during the vulnerability matching because they do not modify the values of the source $id, nor do they affect the execution of the subsequent sink echo.

### 2.2 Our Idea

In summary, injection-based vulnerabilities are usually caused by a lack of protection in two aspects of the source-to-sink path: (1) the source lacks appropriate data-flow sanitization before reaching

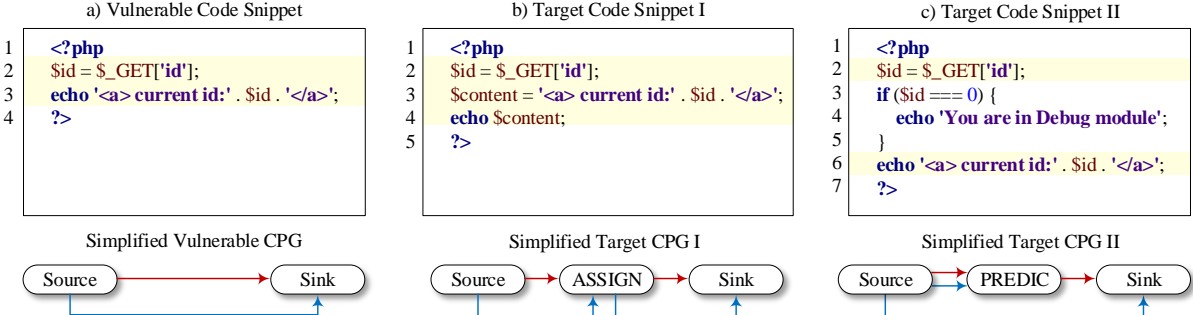

**Figure 1: Three code snippets with exactly consistent vulnerability semantics. Note: lines on light yellow background represent vulnerable code; the red arrows represent control-flow edges; the blue arrows represent data-flow edges;**

the sink; (2) the sink lacks sufficient control-flow constraints before execution. Thus, precisely representing the cause of known vulnerabilities requires considering both the data flow information of the sink parameters and important control flow constraints before the sink. In addition, to address the limitations of HiddenCPG, we aim to represent and match known vulnerabilities in a way that is tolerant to code changes. However, there are two main challenges to achieving this.

- **Challenge-I:** *Different implementations for the same data flow.* As mentioned earlier, the data flow of sink parameters on the source-to-sink path is an important factor in modeling vulnerability causes. Many vulnerability detection approaches [22, 26, 44] leverage graph-based structures (e.g., PDG, CPG, DDG) to represent the data flow information. However, as discussed in §2.1, such graph-based representation is sensitive to the implementations of data flow. That is, different implementations for the same data flow correspond to different graph-based representations, which leads to FN in recurring vulnerability detection. As illustrated in Figure 1 (a) and (b), although the value of the sink parameter in the target program remains unchanged, HiddenCPG generates different CPGs for the two pieces of code due to the newly introduced assignment (line 3 in Figure 1 (b)), resulting in missing the vulnerability. Therefore we need to introduce a new representation of data flow that will not be affected by different implementations.

- **Challenge-II:** *Irrelevant control-flow statements.* The control flow constraint on the sink-to-source path is another important factor to consider when detecting recurring vulnerabilities. Though the data flow on the sink-to-source path within the target program is the same as that of a known vulnerability, it may have been safeguarded by control-flow constraints (such as preventing path traversal vulnerabilities by checking for illegal characters ".." or "/" in the filename or path.). However, there might be many new control flow statements around the copied vulnerable code, most of which are irrelevant to the vulnerability cause. Thus taking all control-flow statements along the source-to-sink path in the target program for matching recurring vulnerabilities will introduce a lot of missing reports. In a word, filtering irrelevant control flow statements while preserving security-relevant control flow constraints is challenging for vulnerability matching.

**Solution:** To address the above challenges, we introduce two new techniques. First, inspired by previous works [35, 46, 47], we utilize symbolic tracking techniques to precisely represent the data flow of known vulnerabilities and introduce **symbolic vulnerable data flow matching** to detect recurring vulnerabilities. This representation and matching scheme is not affected by different implementations of the same data flow. Second, in light of the observation that the control flow constraints introduced by the security patches are key to blocking the vulnerability, we propose **selective safe control flow checking** to check whether the vulnerable code in the target program is blocked by control flow constraints. The checking extracts critical constraints from the patched version, so it is not affected by vulnerability-independent control-flow statements. Specifically, our approach works on two main fronts:

❶ *Symbolic vulnerable data flow matching.* To simultaneously achieve precision in representation and tolerance for differences, our approach leverages forward symbolic tracking to calculate the range of values when the source reaches the sink, represented by expressions composed of various symbols. When a similar source-to-sink path is identified in the target application, our approach computes symbolic expressions for both the sink of known vulnerability and that of the target application and then determines whether they match by computing the similarity of the symbolic expressions for each sink parameter. If the symbolic expressions for each sink parameter match, it means that the data flow cause of the known vulnerability is present in the target program, indicating the potential presence of a recurring vulnerability.

❷ *Selective safe control flow checking.* After matching a potential vulnerability, we further inspect the control flow constraints in the target program to determine whether the recurring vulnerability has been fixed. To filter out the irrelevant control flow statements, we propose a selective approach that centers on the patching behaviors of known vulnerabilities. In general, control-flow constraints newly introduced in security patches are designed to fix known vulnerabilities. Our approach extracts these constraints by analyzing patch modifications. During vulnerability detection, when such constraints manifest on the source-to-sink paths in the target program, it suggests that the path may remain unaffected by the known vulnerability, and vice versa, it suggests that a recurring vulnerability exists.

In summary, if there exists a path within the target program that shares similar vulnerable expressions with known vulnerabilities and lacks any safe constraints introduced in security patches, our approach will report it as a recurring vulnerability.

## 2.3 Running Example

We use a real-world example to further illustrate how RECURSCAN accurately detects recurring vulnerabilities using known vulnerability patches. Figure 2 depicts the overall workflow of RECURSCAN. **The Vulnerability:** The input to RECURSCAN is a security patch for CVE-2019-14530 (Figure 2 (a)), an arbitrary file read vulnerability reported in OpenEMR 5.0.2. The root cause of the vulnerability is that the parameter ($finalZip) of readfile() (line 11) can be controlled by an attacker, allowing directory traversal (e.g., "../../") for reading arbitrary files. The developers fixed the vulnerability by introducing a new control-flow constraint to check whether the user input file name contains illegal characters (lines 7-9).

**Signature Generation:** For the given patch, RECURSCAN employs a four-step process to construct signatures for recurring vulnerability detection. First, RECURSCAN identifies the vulnerability contexts by employing various static analysis techniques on the patch modifications, such as forward taint propagation and backward slicing. This initial step effectively removes vulnerability-irrelevant code, ensuring the accuracy of vulnerability signature generation in subsequent steps (Figure 2 (b)). Second, RECURSCAN performs forward symbolic tracking along the source-to-sink path to calculate the symbolic expression for each sink parameter ($finalZip). The step can represent $finalZip as a clear expression, thereby mitigating the impact of code differences in data flow assignments for subsequent matching. Third, RECURSCAN collects the newly introduced condition statements as control flow constraints by analyzing the differences in the vulnerability context between the vulnerable and patched versions. Finally, RECURSCAN normalizes the data-flow expression and control-flow constraint to derive the vulnerability signature (Figure 2 (c)).

**Vulnerability Detection:** Regarding the recurring vulnerability detection, RECURSCAN primarily focuses on sinks of the same type as the known vulnerability in the target program (e.g., readfile() and unlink() for arbitrary file operations). In all, RECURSCAN leverages a four-step approach to detect recurring vulnerabilities for these potential target sinks. First, RECURSCAN excludes target sinks whose parameters are constants or whose number of parameters do not match the known vulnerability sinks. Second, for the remaining target sinks, RECURSCAN performs backward slicing on its parameters to locate the sources and further slices the interested code contexts of the target sinks. Third, RECURSCAN performs forward symbolic tracking to calculate the data-flow expressions and control-flow constraints among these interested code contexts. Finally, RECURSCAN detects recurring vulnerability by checking whether the expressions and constraints of the target sinks satisfy the following two conditions: (1) target expressions are sufficiently similar to the vulnerable expressions in the signatures; (2) target constraints do not include any safe constraints from the same vulnerability type within the signatures. As shown in Figure 2 (d), RECURSCAN successfully matched the similar code in lh-ehr using the vulnerable expression from the known vulnerability in OpenEMR. However, since the target program already includes the safe

control-flow constraint (line 3) that provides the same protection as the patch introduced, RECURSCAN does not report this case as a vulnerability. On an older version (before v4.1.0) of lh-ehr, the target sink was not protected by the constraint and our approach successfully detects the recurring vulnerability.

## 3 RECURSCAN

In this section, we provide a detailed description of how RECURSCAN works. Overall, RECURSCAN consists of the three key modules. The *Vulnerability Context Slicing (§3.1)* module slices the code related to the cause of known vulnerabilities as the vulnerability context by analyzing the patch modifications. The *Vulnerability Signature Generation (§3.2)* module constructs the signature for each vulnerability by calculating the data-flow expressions from the vulnerable version and extracting the newly-introduced control-flow constraints from the patched version. The module helps to construct a vulnerability signature database from a large number of known vulnerabilities. The *Recurring Vulnerability Detection (§3.3)* module leverages the signature database to accurately search unknown recurring vulnerabilities in target applications.

## 3.1 Vulnerability Context Slicing

First of all, RECURSCAN identifies the code contexts of known vulnerabilities and slices related code for later analysis. In particular, the slicing consists of the following three steps.

**Step I: Patch Context Analyzing.** Since the security patches tend to enhance the protection of sinks around the vulnerable code, RECURSCAN first locates the patch context for further vulnerability context analysis. Particularly, RECURSCAN performs taint propagation from the patch modification lines and collects all tainted statements as patch context. RECURSCAN propagates the taint iteratively from two aspects:

- *Data-flow propagation:* Security patches typically sanitize the source (i.e., untrusted input) to fix vulnerabilities, implying that the variables modified by patches might be related to the cause of the vulnerability. Therefore, RECURSCAN taints all variables in patch modifications and collects all their usage through data dependency analysis.
- *Control-flow propagation:* Another common patching behavior is to introduce conditional statements to interrupt the malicious execution in advance (e.g., lines 6-9 in Figure 2 (a)), thus protecting sensitive sinks in subsequent code. As a result, RECURSCAN taints the statements guarded by conditional statements introduced by the patch through control flow analysis.

**Step II: Source & Sink Locating.** RECURSCAN proceeds to identify the sources and sinks of known vulnerabilities from the patch context. Drawing on previous work that models various PHP built-in sinks for different vulnerability types [11], RECURSCAN employs the same approach to identify the sinks among the patch context code. Then, for each sink, RECURSCAN performs backward slicing on its parameter until any one of the three types of variables is reached, i.e., global variables, super-global variables, and constants. These identified variables are then considered the sources of the known vulnerabilities.

**Figure 2: The running example of how RecurScan leverages security patches to detect recurring vulnerabilities. Note: the lines on light yellow background represent vulnerable code; the lines on light green background represent security constraint.**

**Step III: Vulnerability Context Slicing.** Finally, RecurScan slices the context code of known vulnerability by performing forward analysis from the identified sources to sinks. During this process, RecurScan pays attention to two types of statements: assignment statements and control-flow statements. RecurScan takes such statements into the vulnerability context only if they have direct or indirect data-flow dependencies on the sources. For example, as illustrated in Figure 2, for the given patches, RecurScan identifies lines 2, 5, 6, 10, and 11 as the vulnerability context in the vulnerable version.

In addition, although the patch context analysis has already excluded much vulnerability-irrelevant code, the sink localization in step II may still introduce false positives due to the conservatism of static analysis. To eliminate such incorrect sinks, RecurScan refers to the patched version. In particular, since the security patches only enhance protection for real vulnerability sinks, the vulnerability contexts sliced from the correct sink on the vulnerable version and the patched version must be different. Therefore, for each pair of sink and source, RecurScan performs slicing on both vulnerable and patched versions and eliminates false sinks by comparing whether the two slices are different.

## 3.2 Vulnerability Signature Generation

Following the idea presented in §2.2, RecurScan generates a signature for each known vulnerability by modeling both the data flow expression and control flow constraints. In particular, the signature generation consists of the following three steps.

**Step I: Symbolic Expression Calculating.** As introduced in §2.2, RecurScan aims to represent the vulnerable data flow of the known vulnerability (i.e., the data flow of each sink parameter) with symbolic expressions. To calculate the symbolic expression for the sink parameters, RecurScan performs symbolic tracking along the source-to-sink path contained in the vulnerability context. More concretely, RecurScan follows the control-flow edges from the source-to-sink path. Once reaching an assignment statement, RecurScan iteratively propagates the symbolic values to represent the assigned variables through data dependency analysis. Finally, upon reaching the sink, RecurScan calculates the symbolic expression that represents the value range for each sink parameter. Finally, the symbolic expression only consists of the function name

and three types of symbolic variables, which are global variables, super-global variables, and constants (i.e., the sources of the vulnerability). Thus, this representation method could tolerate certain code changes (e.g., different assignment processes as shown in Figure 1 (a) and (b)) between two versions of the vulnerable code.

**Step II: Safe Constraint Extracting.** To accurately model the control flow constraints of a vulnerability, RecurScan only extracts the safe control flow constraints used for security protection along the source-to-sink path while filtering out the irrelevant ones. The insight of RecurScan's extracting is that only the constraints newly introduced by security patches are specifically intended to fix the vulnerabilities. In particular, RecurScan compares the vulnerability context extracted from vulnerable and patched versions and locates the control-flow statements introduced by the patch. For these newly introduced control-flow statements, RecurScan collects their conditional statements as safe control-flow constraints. For instance, as shown in Figure 2 (a), RecurScan considers only line 7 as a control-flow constraint and disregards line 6.

**Step III: Signature Generation.** After obtaining the symbolic expression and safe constraints, RecurScan generates the vulnerability signature through normalization. In particular, for the symbolic expressions, RecurScan normalizes its symbolic variables while retaining the function names. Specifically, RecurScan normalizes global variables to $Global, super-global variables to $Source, and constants to Constant. Regarding the control-flow constraints, RecurScan only normalizes the variables in conditional statements with $var while preserving constant strings, as they may represent regular expressions or black/white lists used for security protection. Such normalization strengthens the generality of the vulnerability signatures, making them tolerant of code changes such as variable renaming.

**Signature Database Construction:** Following the signature generation method, RecurScan constructs a signature database containing hundreds of known vulnerabilities. To speed up the matching process, RecurScan classifies all normalized expressions and constraints based on their vulnerability types when constructing the database. In addition, RecurScan also removes duplicate expressions and constraints within the same vulnerability type.

## 3.3 Recurring Vulnerability Detection

Considering that the same vulnerability type involves similar sink functions, RECURSCAN analyzes the target program for each vulnerability type and detects recurring vulnerabilities belonging to that type. For each vulnerability type, RECURSCAN first locates all the potential sinks within the target applications (e.g., `echo`, `print` or `printf` for XSS vulnerability) and employs backward slicing to locate the corresponding sources. Then, RECURSCAN employs a similar process as introduced in §3.2 to calculate the data-flow expressions ($D_{\texttt{target}}$) and control-flow constraints ($C_{\texttt{target}}$) for each source-to-sink path. Finally, RECURSCAN tries to match them with the vulnerability signatures (i.e., the symbolic expressions $D_{\texttt{vuln}}$ and the safe constraints $C_{\texttt{safe}}$) of the same type of vulnerabilities in the signature database.

In particular, RECURSCAN only reports a recurring vulnerability if there is a known vulnerability such that the target source-to-sink path meets the following criteria: (1) For each sink parameter, the text similarity between the target expression $D_{\texttt{target}}$ and vulnerability expression $D_{\texttt{vuln}}$ is greater than a threshold $T$. (2) The control-flow constraints $C_{\texttt{target}}$ does not contain any safe constraints of $C_{\texttt{safe}}$. The $T$ represents a predefined similarity threshold, which enables RECURSCAN to be resilient to code differences in the target applications while maintaining its accuracy. In this way, RECURSCAN can precisely match potential recurring vulnerabilities while also tolerating code differences.

## 4 EVALUATION

### 4.1 Experimental Setup

**Prototype.** We implemented a prototype of RECURSCAN, which consists of 5,275 lines of Python code. The prototype is built upon PHPJoern [11] and performs various static program analyses using CPG queries on Neo4j [10]. The *Vulnerability Context Slicing* module utilizes the `GitPython` library to identify patch modifications on GitHub [9]. In the *Recurring Vulnerability Detection* model, we have set the similarity threshold to 0.95 to allow for precise matching while also tolerating code differences.

**Experiments.** Our evaluation is organized by answering the following research questions:

- RQ1: How effective is RECURSCAN in automatically generating vulnerability signatures?
- RQ2: How effective is RECURSCAN in detecting recurring vulnerabilities in real-world applications?
- RQ3: How accurate is RECURSCAN compared to the state-of-the-art approaches?
- RQ4: How efficient is RECURSCAN in performing the end-to-end analysis?

**Known Vulnerability Dataset.** RECURSCAN requires known vulnerability patches as input to automatically construct the signature database for recurring vulnerability detection. To maintain the quality of the patch collection, we follow two criteria: (1) the known vulnerability should be of the injection-based type and have been disclosed within the past 5 years, with corresponding security patches available; (2) the application in which the vulnerability is discovered should be implemented in PHP and possess a sufficient level of popularity, defined as having more than 100 stars on

**Table 1: The overview of signature database.**

| Type Classification | Vuln Expressions | Safe Constraints |
|---|---|---|
| Malicious Code Injection[1] | 221 | 6 |
| Arbitrary File Operation[2] | 28 | 21 |
| **Total** | **249** | **27** |

[1] Including XSS, SQL injection, command injection, etc.
[2] Including arbitrary file inclusion, upload, write, read, and delete.

GitHub [9]. Specifically, we use a crawler with several keywords (e.g., "PHP", "XSS" or "SQL") to search for CVEs from the NVD (i.e., National Vulnerability Database) [8]. As a result, we successfully collect 228 CVEs that satisfy our criteria. Then, we manually filtered out 44 CVEs for their patches that unmet the requirements of subsequent analysis, including 31 CVEs whose patches fix the vulnerability by modifying non-PHP files and 13 CVEs for which PHPJoern failed to parse their patches. In all, our known vulnerability dataset consists of 184 known vulnerability patches from 42 widely-used PHP applications.

**Testing Dataset.** Regarding the testing set, we collect the latest version of 200 PHP web applications from GitHub. The selection criteria are also that all of these applications should be sufficiently popular, i.e., having at least 100 stars on GitHub. In all, the testing set includes 92,499 PHP files and 15,334,595 line of codes.

**Baselines.** Given the high relevance to our work, we compare the effectiveness and efficiency of RECURSCAN with HiddenCPG [6] in detecting recurring vulnerabilities.

**Environment.** The experiments are run on a Ubuntu 20.04 machine with an Intel Xeon Gold 6242 processor and 245 GB memory.

### 4.2 Vulnerability Signatures (RQ1)

In this experiment, we break down the signature database automatically constructed by RECURSCAN. Table 1 presents the overall results. In all, RECURSCAN successfully calculates 249 vulnerable data-flow expressions and extracts 27 safe control-flow constraints from 173 known vulnerability patches. Notably, the patches of the arbitrary file operation vulnerabilities introduce more safe constraints. This aligns with the common practice, where developers tend to fix such vulnerabilities by implementing constraints for filenames or path checking, while preferring to employ data-flow sanitization for handling malicious injection vulnerabilities.

However, for the 11 patches that RECURSCAN failed to analyze, we conducted a thorough investigation and found that all of them shared the same root cause: *the information provided by the patch modifications was insufficient to help identify the sinks of vulnerabilities*. The details are as follows:

- *Global variables modification (7 patches).* Patches in this category fixed vulnerabilities by modifying the assignment of global variables. However, since these variables can be used anywhere in the program without data-flow edges in CPGs provided by PHPJoern. This made it difficult for RECURSCAN to identify their patch contexts and sinks based on these limited modifications.
- *Class fields modification (4 patches).* Some patches fixed vulnerabilities by modifying the values of class fields. Due to the inherent challenges in the class def-use analysis, RECURSCAN was unable to identify the sinks for these patches.

Table 2: Breakdown of the detected vulnerabilities.

| Baseline | Type-1 | Type-2 | Type-3 | Type-4 | Total |
|----------|--------|--------|--------|--------|-------|
| RecurScan | 0 | 122 | 106 | 4 | 232 |
| HiddenCPG | 0 | 122 | 2 | 0 | 124 |

## 4.3 Vulnerabilities Detected (RQ2)

We apply RecurScan to the testing set with the constructed signature database. In all, RecurScan reported 251 distinct potential vulnerabilities. Next, we will discuss the quality of these reports.

**Report Verification.** First, we manually investigated the identified vulnerabilities to confirm their exploitability. Given that this process requires significant efforts, three authors have participated, each with a minimum of 3 years of expertise in web security. For each vulnerability report, the analyst will inspect and confirm its exploitability by writing a PoC. Overall, we confirmed that 232/251 (92.43%) reports are indeed vulnerabilities, including 171 XSS, 55 SQLi, and 6 arbitrary file operations. The attackers can exploit these vulnerabilities to compromise the corresponding applications, including initiating denial-of-service (DoS) attacks, stealing confidential database records, and even uploading malicious PHP webshell files to enable remote code execution. As of now, we have reported these severe vulnerabilities to developers and received 89 CVEs.

Regarding the remaining 19 false positives, we break down the reasons as follows: (1) 14 false positives were found in the dead code of the target applications. Although these cases indeed exist recurring vulnerabilities, they were not exploitable for being inaccessible in a running system; (2) 4 false positives that could only be exploited under specific configuration (e.g., debug mode); (3) 1 false positive occurred where the developer modified the HTTP request headers using the `header()` function, causing the access to the vulnerable page to be treated as a file download, which in turn prevented the exploitation of the vulnerability. In essence, all of these false positives stem from the inherent challenges of static analysis and may only be removed via dynamic analysis.

**Clone Types.** Clone type [1, 24, 26, 44, 45] is an important metric used to evaluate the ability of an approach to tolerate code differences when detecting recurring vulnerabilities. In general, it categorizes recurring vulnerabilities into the following four types based on their code differences compared to known vulnerabilities:

- *Type 1:* Exact copy, only differences in white space and comments.
- *Type 2:* Same as type 1, but also variable renaming.
- *Type 3:* Same as type 2, but changing or adding a few statements.
- *Type 4:* Semantically identical, but not necessarily same syntax.

Therefore, we further analyzed the clone types of vulnerabilities detected by RecurScan. Table 2 presents the clone types of identified recurring vulnerabilities. The results demonstrate that RecurScan effectively identifies vulnerabilities that fall into Type-2, Type-3, and Type-4. The absence of Type-1 lies in RecurScan both constructs signatures and detects vulnerabilities in real-world applications. The source code of these applications indeed exhibits certain syntax differences. However, we believe that if Type-1 were to occur, RecurScan would easily identify them based on its excellent performance in detecting instances of other types.

Note that detecting Type-3 and Type-4 vulnerabilities is known for its significant technical challenges [19, 24, 44, 45]. RecurScan

Table 3: Effectiveness comparison.

| Baseline | TP | FP | FN | Prec(%) | Recall(%) |
|----------|----|----|----|---------|-----------|
| RecurScan | 232 | 19 | 0 | 92.43 | 100.00 |
| HiddenCPG | 124 | 45 | 108 | 73.37 | 53.45 |

successfully detected 106 Type-3 clones and 4 Type-4 clones. This achievement can be attributed to the symbolic and selective comparison method, which enables RecurScan to tolerate the code changes while precisely matching the recurring vulnerabilities.

## 4.4 Comparison (RQ3)

In this experiment, we compared the effectiveness of RecurScan and HiddenCPG in detecting recurring vulnerabilities. To ensure fairness, both of them are applied to the testing set using the same known vulnerability dataset. We compared their accuracy with two metrics: precision and recall.

**Ground Truth.** Comparing the accuracy of each work requires a comprehensive enumeration of all vulnerabilities within the testing set, which is infeasible. Therefore, to ensure a fair comparison, we construct a ground truth by aggregating all vulnerabilities detected by both RecurScan and HiddenCPG in our testing set. Note that each vulnerability involved in the ground truth was meticulously examined by manually analyzing reported potential vulnerabilities and confirming them as true positives. In total, the ground truth consists of 232 vulnerabilities. It is worth noting that RecurScan can detect all of these cases, showcasing its remarkable capability in detecting recurring vulnerabilities.

**HiddenCPG Setup.** We follow the guidance and run HiddenCPG with three steps. First, we use the command "`python mkcpg.py  <CPG>`" to convert the code of known vulnerabilities and target programs into CPGs. Second, we use the command "`python Extractor.py <CPG_vuln> <vulnerable path>`" to eliminate irrelevant nodes from vulnerable CPGs. Finally, we leverage HiddenCPG to identify recurring vulnerabilities by using the command: "`python hiddencpg.py <CPG_target> <CPG_vuln>`".

**Results Overview.** The comparison results between RecurScan and HiddenCPG are presented in Table 3. Overall, RecurScan outperforms HiddenCPG by 25.98% in precision and 87.09% in recall. Within the ground truth consisting of 232 vulnerabilities, RecurScan accurately identifies all of them and reports only 19 false positives. While HiddenCPG, limited by strict graph matching, detected only 124 true positives but also reported 45 false positives. This clearly demonstrates the advantages of RecurScan in accurately detecting recurring vulnerabilities.

**False Positive Analysis.** Given that HiddenCPG works on strict graph matching, we were surprised by the high number of false positives it reported, specifically 45 cases. After a rigorous analysis, we identified that apart from 19 cases also reported by RecurScan, HiddenCPG additionally detected 26 more false positives. The main reason for this is that HiddenCPG uses the CPG of known vulnerable code `print_r($_GET["a"])` for isomorphic subgraph matching. As a result, HiddenCPG could match many target codes like `print_r($_GET["a"], true)` and report them as potential XSS vulnerabilities. In reality, though the code property graphs of these target codes include the isomorphic vulnerable subgraphs, they are not exploitable by attackers. The reason lies in that `print_r`

function has a "return" parameter, and when set to "true", the function returns the output `$_GET["a"]` as a string instead of printing it, rendering it non-exploitable by attackers. Note that RecurScan does not report these false positives because it compares each expression of the sink parameters in target applications with known vulnerabilities.

**False Negative Analysis.** For 108 false negatives, we provide the reasons why HiddenCPG failed to detect them. As illustrated in Table 3, these false negatives consist of 104 Type-3 and 4 Type-4 clones, exhibiting certain code differences compared to known vulnerabilities. Blamed for the low tolerance to differences in subgraph matching, HiddenCPG is unable to detect these vulnerabilities. In contrast, RecurScan will not suffer from these false negatives thanks to the symbolic and selective comparison method.

### 4.5 Efficiency (RQ4)

We evaluated the efficiency of RecurScan across the entire testing set. In total, RecurScan took approximately 20 days and 14 hours to complete the task of vulnerability detection in 200 PHP web applications. That is, each application took about 2.5 hours to analyze on average. The primary time cost was attributed to the analysis of symbolic tracking. More specifically, RecurScan needs to calculate the symbolic expressions for all the potential sink functions in the target program. However, such a heavy and accurate analysis also enables RecurScan to tolerate the code differences in the target applications, which finally achieves a superior performance in detecting Type-3&4 recurring vulnerabilities.

In contrast, HiddenCPG completed the vulnerability detection on the entire testing set in about 12 days and 2 hours (i.e., 1.5 hours per application). Although HiddenCPG is a little more efficient than RecurScan, RecurScan detects significantly more vulnerabilities than HiddenCPG. Considering that vulnerability detection is usually an offline task, we believe a stronger capability in vulnerability detection is more favorable.

### 5 DISCUSSION

**Trade-off.** To balance the pros and cons of complex control flows in recurring vulnerability matching, we choose a trade-off approach. Nevertheless, in the case of limited patches set, the constraints automatically extracted by RecurScan may not be comprehensive enough to cover all security measures, potentially leading to false positives in vulnerability detection. However, this limitation can be addressed in the future as the scale of patches continues to expand.

**Vulnerability Scope.** Although the prototype of RecurScan targets injection-based vulnerabilities, we argue that it could also be applied to other types of vulnerabilities. The workflow of RecurScan revolves around the sources and sinks, which are most commonly found in the causes of injection-based vulnerabilities but also occur in some cases of broken access control. There have been several existing work [30, 38] that model security-sensitive operations as sinks (e.g., `mysql_query`) and analyze whether the sources are properly checked when reaching these sinks. Such types of source-to-sink vulnerabilities could also be included in our analysis scope.

**Generalization.** Our prototype of RecurScan is designed for PHP applications. Nonetheless, note that source-to-sink vulnerabilities can also arise in non-PHP web applications (e.g., Java and Node.js).

Therefore, the general approach for detecting recurring vulnerabilities can be applied seamlessly to these systems. For the adaptation, the end-users could re-implement all three modules with the corresponding static analysis frameworks (e.g., Soot [41] for Java).

### 6 RELATED WORK

**Web Vulnerabilities Detection.** In recent years, numerous techniques have been proposed to automatically detect vulnerabilities within web applications. A commonly used technique is static analysis [11, 13–15, 20, 27, 29, 31, 37, 39, 42], but it relies on expert-level programming to represent various vulnerability patterns, which is labor-intensive and prone to errors. Another well-known technique is dynamic analysis [16–18, 32, 33, 36, 40], which employs crawling and fuzzing techniques to identify web vulnerability in a black-box fashion. However, these approaches are often limited by code coverage, which can result in many false negatives.

**Vulnerable Code Clone Detection.** There are a lot of existing work focused on identifying recurring vulnerabilities by vulnerable code clone detection. Zhou *et al.* [28] proposed CP-Miner to detect bugs caused by inconsistent identifier renaming in code clones. Jang *et al.* [19] introduced a token-based approach called ReDeBug to locate unpatched code clones at the line-level granularity. Kim *et al.* [24] presented VUDDY, a scalable approach for detecting vulnerable code clones using several vulnerability-preserving abstraction techniques. However, these works are primarily suited for detecting Type-1 or Type-2 clones. While for Type-3 clones, Li *et al.* [26] and Wi *et al.* [44] introduced CBCD and HiddenCPG, respectively. These approaches detect vulnerabilities by solving a subgraph isomorphism problem. However, due to the inherent challenges of subgraph matching, both of them still face numerous false positives and false negatives when matching the vulnerabilities. Xiao *et al.* [45] proposed MVP, like RecurScan, takes security patches as input and focuses on the differences between the vulnerable and patched versions. However, its scope is limited by function-level analysis, making it difficult to apply for inter-procedural detection. The last one is Tracer [21], which is equipped with an existing static analyzer [12]. To detect vulnerable code at the semantic level, Tracer designs a series of low- and high-level features to represent the characteristics of known vulnerabilities. However, as it is tailored for C-based programs, adapting it for PHP web applications can be less effective. Unlike these works, thanks to the symbolic tracking and selective matching techniques, RecurScan can excel in Type-3 but also some Type-4 clone detection.

### 7 CONCLUSION

In this paper, we proposed RecurScan, a novel approach designed for accurately detecting recurring vulnerabilities with resilience to code differences. RecurScan centers around security patches and symbolic tracking techniques, matching recurring vulnerabilities by comparing symbolic expressions and selective constraints with known vulnerable code. This approach proves effective in tolerating code differences while achieving precise matching results. Overall, RecurScan successfully identified 232 recurring vulnerabilities among 200 popular PHP applications, outperforming the state-of-the-art approach by 25.98% in precision and 87.09% in the recall.

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
