# OpenReview forum: "RecurScan: Detecting Recurring Vulnerabilities in PHP Web Applications"
_ACM.org/TheWebConf/2024/Conference — TheWebConf24 Oral_

### Official Review · Reviewer_G5bU · 2023-11-23

**Novelty:** 5
**Technical Quality:** 7

**Review:**

Pros:
- significant empirical improvement over prior work
- demonstrated that it is useful in practice by collecting a sizable amount of previously unknown CVEs
- The paper's technical artifact is also a large technical contribution for program analysis.
- Combining symbolic information with subgraph similarity is a novel technique

The paper presents a method for identifying recurring vulnerabilities in code using symbolic data flow to improve on previous techniques. In this work, they combine prior approach's use of subgraph similarity with symbolic data flow information. The symbolic data flow information helps the tool to identify similar bugs under different conditions by checking for similar data flows.

The paper's description of why RecurScan is able to detect things that HiddenCPG is very abstract. Concrete examples here would help to give more examples about why the new system is capable of identifying more vulnerabilities.

**Questions:**

Questions:
- How generalizable are the techniques used in this paper? Would they apply to other languages for example?
- Why does RecurScan detect things that HiddenCPG doesn't? What is the insight to why the new method works?
- What avenues for future work are there in this area?

**Reviewer Confidence:**

3: The reviewer is confident but not certain that the evaluation is correct

**Scope:**

4: The work is relevant to the Web and to the track, and is of broad interest to the community

---

### Official Review · Reviewer_RnR5 · 2023-11-23

**Novelty:** 6
**Technical Quality:** 6

**Review:**

### Summary

The paper presents a novel approach for detecting previously unknown vulnerabilities in PHP code. This method involves generating vulnerability signatures from existing security patches. These signatures are then analyzed through both data and control flow analysis, enabling the extraction of critical control-flow constraints and data-flow expressions. Leveraging this information, the proposed tool, RecurScan, creates refined vulnerability signatures. These signatures allow identifying similar vulnerabilities, even in presence of code variations. Employing this technique, the authors successfully uncovered 232 previously unknown vulnerabilities. Additionally, they demonstrate that an existing state-of-the-art method, HiddenCPG, only managed to detect a fraction of these vulnerabilities due to its inability to effectively handle variations in code.


### Strengths

+ Well-written
+ Good methodology
+ Good evaluation in real-world setting
+ Identification of large number of unknown vulnerabilities

### Weaknesses

- Limited scope (PHP, injection vulnerabilities)


### Detailed comments

The paper is well-written and includes helpful examples that enable readers to easily follow the descriptions provided by the authors. The core idea of the approach is to obtain more generic signatures capable of identifying common vulnerability patterns, even in altered code. The effectiveness of this proposed approach is supported by a thorough evaluation on real codebases, where the authors identified 232 vulnerabilities. Impressively, they received CVEs for 89 of these vulnerabilities. The paper also discusses false positives and compares their method with another state-of-the-art approach, HiddenCPG. In this comparison, RecurScan not only detects all vulnerabilities identified by HiddenCPG but also additional ones, particularly those exhibiting more significant code differences from the training data. Furthermore, the efficiency of both approaches is compared.

Overall, RecurScan appears to significantly advance the state of the art, and the authors present impressive results. The evaluation is sound and comprehensive. Although it has been tested only on PHP code, the approach should be transferable to other programming languages as well.


#### Typos

- line 79: anlaysis $\rightarrow$ analysis
- line 194: exactly same $\rightarrow$ exact same
- line 483: known vulnerability $\rightarrow$ of a known

**Questions:**

- In the given example, both, the sink and the source are part of the available patch. However, in practice we can also have patches that do not contain the source/sink in a given patch. Can RecurScan also handle these cases? If no, the authors should briefly discuss this limitation in Section 5.
- The authors exclude sinks which parameters do not exactly the number of those of their examples with vulnerable code. How do they ensure that they do not miss potential (unknown) vulnerabilities? Or is this a trade-off they had to make for performance reasons?
- Do the authors have plans to release the code and data to the public (eventually)?
- Have the discovered security vulnerabilities been fixed, or do they still pose a risk?

**Ethics Review Description:**

Authors responsibly reported found vulnerabilities to developers, so I don't see any ethical issues with this work.

**Reviewer Confidence:**

3: The reviewer is confident but not certain that the evaluation is correct

**Scope:**

3: The work is somewhat relevant to the Web and to the track, and is of narrow interest to a sub-community

---

### Official Review · Reviewer_B8Vc · 2023-11-23

**Novelty:** 4
**Technical Quality:** 7

**Review:**

Dear authors,

thank you for submitting to WWW. I found your paper an interesting read and would like to share some comments below, which i hope will help you in strengthening the papers' presentation.

# In-Class generalization

Given that the implementation ultimately created abstractions of ingested vulnerabilities, it would be a relevant data-point to see how well these abstractions generalize, i.e., if there is a consistent set of signatures that encompases all (or at least, say, 80%) vulnerabilities. This datapoint is relevant to better understand the impact of the ingestion and analysis phase of the proposed implementation.

This in-class generalization also ties in with the handling of semi-constant control flow aspects, e.g., the the exact regular expression in Figure 2. There, I would argue that the proposed system should not have identified a case where the regex was adjusted to be `[^A-z0-9_.-]`. I am looking forward to feedback from the authors on whether i correctly understood this aspect.

# Related work from usable security

There is a relatively large body of related work regarding code re-use and its security implications. The paper currently does not engage with that body of related work, even though it likely is very useful in motivating some of the currently uncited (but accepted as true by this reviewer given the state of the literature). As a starting point, i would recommend: https://ieeexplore.ieee.org/abstract/document/7546508

# Random Thoughts

This is not a comment but a random thought on the motivation of the work. There is no need to 'address' this; Reading the motivation of the paper, i was--to a degree--struck by the relevance of the work to Copilot (and similar code generation systems). Particularly, these systems should introduce similar vulnerabilities for reasons comparable to why humans do it (well: essentially copy-pasting). Similarly, it might be a fun study to generate a lot of web-app code in PHP with co-pilot, and use the presented framework to assess the produced code.

# Summary
Overall, I believe that this paper highlights a good application of ongoing research to a problem very relevant to web-security. Especially the practical impact in terms of detected and mitigated vulnerabilities is impressive. I remain somewhat cautious concerning the novelty of some concepts due to my limited familiarity with the fuzzing/control flow analysis field. My review should hence be understood under that constraint of mine.

**Questions:**

- Would you please comment on the point about adjusted control-flow constraints above, i.e., whether I missed something there

**Ethics Review Description:**

-

**Reviewer Confidence:**

2: The reviewer is willing to defend the evaluation, but it is likely that the reviewer did not understand parts of the paper

**Scope:**

4: The work is relevant to the Web and to the track, and is of broad interest to the community

---

### Official Review · Reviewer_LQNi · 2023-11-24

**Novelty:** 5
**Technical Quality:** 6

**Review:**

The paper introduces RecurScan, a new approach to detecting recurring vulnerabilities in PHP web applications. The authors explain how RecurScan works by using symbolic tracking techniques to match recurring vulnerabilities by comparing symbolic expressions and selective constraints with known vulnerable code. The authors state that RecurScan outperforms the state-of-the-art approach and conclude that RecurScan is a promising tool for detecting vulnerabilities in PHP web applications.


# Strengths

- The paper explains the drawbacks of the SOTA work clearly and points out possible solutions to solve challenges like low resilience problem in sub-graph matching and provides concrete examples to show how it fails in different settings.

- The authors are keenly aware of the challenges of using patches for recurring vulnerability identification and elaborate on them, with a broadly correct understanding of the techniques involved, and are able to use existing techniques to enhance the effectiveness of static analysis while achieving good results.

- The methods used by the authors are well thought out and present a solution to the specific security problem of recurring vulnerability detection. The methodology is generally reliable and accurate. In addition, effectively detecting recurring vulnerability in PHP applications will contribute to the community.

- The paper has a clear organization and is easy to follow.

- The authors have found and responsibly reported many new vulnerabilities using their technique.


# Weaknesses

- This work does not address substantial technical challenges and presents limited novel techniques.

The main techniques used in the paper are generally very well developed, e.g., taint analysis or symbolic execution, and there are many related mature solutions in the community. The symbolic expression similarity comparison as well as the safe constraint checking approach used by the authors are quite straightforward to implement, but they indeed help improve the detection ability.

- The comparison with HiddenCPG is not completely fair.

This paper uses HiddenCPG as the baseline to show the good performance of RecurScan. The claim that RecurScan "outperforms the state-of-the-art approach by 25% in precision and 87.09% in recall" is not well supported. The authors use the results reported by RecurScan and HiddenCPG as the ground truth dataset, which is not fair. The authors could have evaluated using the dataset released by the authors of HiddenCPG as the ground truth.

- Many technical details are unclear.

How RecurScan compares the data-flow expressions (by text similarity) and how it propagates taints are unclear.

It is also unclear if the analyses (e.g., patch context analysis, symbolic tracking) are inter-procedural.

It is not described whether and how RecurScan handles built-in functions and inter-procedural calls in the taint analysis. These are known as difficult problems in PHP taint analysis as discussed in TChecker [29]. Without such supports, the taint analysis is just standard and can be very inaccurate.

It is unclear how the symbolic execution (tracking) is performed, how the authors address the challenges such as built-in functions, and how the authors' methods compare to the related work, such as [R1] that the authors fail to discuss.

- Some of the descriptions in the paper are inaccurate or even incorrect.

At the beginning of section 4.5, the authors say "Although HiddenCPG is a little more efficient than RecurScan, RecurScan detects significantly more vulnerabilities than HiddenCPG". In terms of the total time spent analyzing all applications, RecurScan takes nearly twice the time HiddenCPG spent, so how can HiddenCPG be only a little more efficient? Is this statement fair? Also, using more resources for analysis should yield better results, and the sub-graph matching problem is technically more time-consuming than the expression matching problem.

In Figure 1(b), why are there two control flow edges? The fact is that there is only one basic block in that example. Such a representation is technically wrong, so please check it carefully.

The statement in L49-50 also reads wrong. Why do you still consider sanitization of untrusted inputs vulnerable?

The data-flow and control-flow propagation in Step I of 3.1 are unclear and need further clarification or correction. It is described some variables and statements are tainted. But it is unclear how the taints are propagated, to which direction, and to what sinks. The description reads like just standard data-flow analysis, e.g., slicing, instead of taint propagation. Please correct the inaccurate description.

L615: model -> module


[R1] Penghui Li, et al. "On the Feasibility of Automated Built-in Function Modeling for PHP Symbolic Execution". In WWW '21.

**Questions:**

- Can you clarify on the unclear technical details?

- In order to contribute to future research in the security community, do you plan to release the source code of this work as well as the dataset? This would be very helpful for reproducing your results and facilitate future research.

- How was the threshold (0.95) determined?

- RecurScan excludes target sinks whose number of parameters does not match the number of known vulnerability sinks. Would this lead to false negatives?

**Reviewer Confidence:**

4: The reviewer is certain that the evaluation is correct and very familiar with the relevant literature

**Scope:**

4: The work is relevant to the Web and to the track, and is of broad interest to the community

---

### Official Review · Reviewer_SWKb · 2023-11-24

**Novelty:** 5
**Technical Quality:** 5

**Review:**

# Summary:
This paper introduces RecurScan, a novel approach for detecting recurring vulnerabilities in PHP web applications by leveraging security patches. RecurScan circumvents the limitations of traditional static analysis methods, which are often thwarted by minor code variations and require laborious manual modeling of vulnerability causes. By combining symbolic tracking and control-flow analysis, RecurScan generates signatures representing the data-flow and control-flow characteristics of known vulnerabilities. It then detects recurring vulnerabilities by matching these signatures against potential vulnerable points in the target application, accounting for variations in code implementation. RecurScan's effectiveness was demonstrated through an evaluation on popular PHP applications, where it outperformed existing techniques, identifying new vulnerabilities with high accuracy and reducing false positives.

# Strength
+ The idea proposed by the paper is novel
+ Well-motivated research problem
+ Well written paper
+ The experiment is extensive and convincing

# Weakness
- Less discussed limitation
- LLM-based approach could be a short-cut to bypass all the efforts
- Lack of baseline with naive clone detectors

# Evaluation:

As shown in my comments in strength, I like the paper at this stage. I would like to applaud the authors for doing a good job in paper writing.

Nevertheless, I would like to ask the following questions for the authors to response:

The authors choose a graph-based approach as a baseline, I would like to see more baselines regarding different clone detectors. Note that, there is no perfect solution for detecting the code duplications, specially for code semantics. By abstracting the code into token sequence, AST, PDG, CFG, or even metrics, they enjoy their advantages in precision, recall, and runtime overhead. Therefore, it seems to me that it is just a tradeoff in the design. In this regard, the authors might consider to include more baselines (e.g., different clone detectors).

As for semantic extraction, I might encourage the authors to use GPT model families for semantic similarity measurement. If it works, a transformer-based model could be another option.

**Questions:**

See my above comments.

**Reviewer Confidence:**

3: The reviewer is confident but not certain that the evaluation is correct

**Scope:**

3: The work is somewhat relevant to the Web and to the track, and is of narrow interest to a sub-community

---

### Decision · Program_Chairs · 2024-01-22

**Decision:**

Accept (Oral)

**Comment:**

The paper is praised by the reviewers by the well-motivated problem, well throughout and reliable approach, and the impressive effectiveness in finding unknown vulnerabilities in real-world settings. The open source of the artifact will advance the field.

 ---